# Molecular-Clinical Correlation in Pediatric Medulloblastoma: A Cohort Series Study of 52 Cases in Taiwan

**DOI:** 10.3390/cancers12030653

**Published:** 2020-03-11

**Authors:** Kuo-Sheng Wu, Donald Ming-Tak Ho, Shiann-Tarng Jou, Alice L. Yu, Huy Minh Tran, Muh-Lii Liang, Hsin-Hung Chen, Yi-Yen Lee, Yi-Wei Chen, Shih-Chieh Lin, Feng-Chi Chang, Min-Lan Tsai, Yen-Lin Liu, Hsin-Lun Lee, Kevin Li-Chun Hsieh, Wen-Chang Huang, Shian-Ying Sung, Che-Chang Chang, Chun Austin Changou, Kung-Hao Liang, Tsung-Han Hsieh, Yun-Ru Liu, Meng-En Chao, Wan Chen, Shing-Shung Chu, Er-Chieh Cho, Tai-Tong Wong

**Affiliations:** 1Graduate Institute of Clinical Medicine, College of Medicine, Taipei Medical University, Taipei 110, Taiwan; abel1063@gmail.com (K.-S.W.); chaomengen@gmail.com (M.-E.C.); a128098527@gmail.com (W.C.); dog52037@gmail.com (S.-S.C.); 2Department of Pathology and Laboratory Medicine, Taipei Veterans General Hospital, Taipei 112, Taiwan; mtho11728@gmail.com (D.M.-T.H.); diegolin@vghtpe.gov.tw (S.-C.L.); 3Department of Pathology and Laboratory Medicine, Cheng Hsin General Hospital, Taipei 112, Taiwan; 4Department of Pediatrics, National Taiwan University Hospital, College of Medicine, National Taiwan University, Taipei 100, Taiwan; stjou4@gmail.com; 5Institute of Stem Cell and Translational Cancer Research, Chang Gung Memorial Hospital at Linkou and Chang Gung University, Taoyuan 333, Taiwan; ayu@gate.sinica.edu.tw; 6Genomics Research Center, Academia Sinica, Taipei 115, Taiwan; 7International Master/Ph.D. Program in Medicine, College of Medicine, Taipei Medical University, Taipei 110, Taiwan; t.minhhuy@gmail.com; 8Department of Neurosurgery, University of Medicine and Pharmacy at Ho Chi Minh City, Ho Chi Minh 700000, Vietnam; 9Division of Pediatric Neurosurgery, The Neurological Institute, Taipei Veterans General Hospital and School of Medicine, National Yang-Ming University, Taipei 112, Taiwan; liang4617@hotmail.com (M.-L.L.); roberthhchen3@gmail.com (H.-H.C.); yylee62@gmail.com (Y.-Y.L.); 10Department of Radiology, Taipei Veterans General Hospital and School of Medicine, National Yang-Ming University, Taipei 112, Taiwan; chen6074@gmail.com (Y.-W.C.); fcchang@vghtpe.gov.tw (F.-C.C.); 11Department of Pediatrics, College of Medicine, Taipei Medical University Hospital, Taipei Medical University, Taipei 110, Taiwan; minlan456@hotmail.com (M.-L.T.); yll.always@gmail.com (Y.-L.L.); 12Pediatric Brain Tumor Program, Taipei Cancer Center, Taipei Medical University, Taipei 110, Taiwan; kevinh9396@gmail.com; 13Department of Radiation Oncology, College of Medicine, Taipei Medical University Hospital, Taipei Medical University, Taipei 110, Taiwan; b001089024@tmu.edu.tw; 14Department of Medical Imaging, College of Medicine, Taipei Medical University Hospital, Taipei Medical University, Taipei 110, Taiwan; 15Department of Pathology, Wan Fang Hospital, Taipei Medical University, Taipei 110, Taiwan; bluemageh@gmail.com; 16The Ph.D. Program for Translational Medicine, Taipei Medical University, Taipei 110, Taiwan; ssung@tmu.edu.tw (S.-Y.S.); ccchang168@tmu.edu.tw (C.-C.C.); 17The Ph.D. Program for Cancer Biology and Drug Discovery, Center for Translational Medicine, Taipei Medical University, Taipei 110, Taiwan; austinc99@tmu.edu.tw; 18Department of Medical Research, Taipei Veterans General Hospital and College of Bioinformatics, National Yang Ming University, Taipei 112, Taiwan; kunghao@gmail.com; 19Joint Biobank, Office of Human Research, Taipei Medical University, Taipei 110, Taiwan; thhsieh2003@gmail.com (T.-H.H.); d90444002@tmu.edu.tw (Y.-R.L.); 20Department of Clinical Pharmacy, School of Pharmacy, College of Pharmacy, Taipei Medical University, Taipei 110, Taiwan; 21Division of Pediatric Neurosurgery, Department of Neurosurgery, Taipei Medical University Hospital, Taipei Medical University, Taipei 110, Taiwan; 22Neuroscience Research Center, Taipei Medical University Hospital, Taipei 110, Taiwan

**Keywords:** medulloblastoma, molecular–clinical correlation, risk stratification, RNA-Seq, somatic mutations, DNA damage response, genetic predisposition

## Abstract

In 2016, a project was initiated in Taiwan to adopt molecular diagnosis of childhood medulloblastoma (MB). In this study, we aimed to identify a molecular-clinical correlation and somatic mutation for exploring risk-adapted treatment, drug targets, and potential genetic predisposition. In total, 52 frozen tumor tissues of childhood MBs were collected. RNA sequencing (RNA-Seq) and DNA methylation array data were generated. Molecular subgrouping and clinical correlation analysis were performed. An adjusted Heidelberg risk stratification scheme was defined for updated clinical risk stratification. We selected 51 genes for somatic variant calling using RNA-Seq data. Relevant clinical findings were defined. Potential drug targets and genetic predispositions were explored. Four core molecular subgroups (WNT, SHH, Group 3, and Group 4) were identified. Genetic backgrounds of metastasis at diagnosis and extent of tumor resection were observed. The adjusted Heidelberg scheme showed its applicability. Potential drug targets were detected in the pathways of DNA damage response. Among the 10 patients with SHH MBs analyzed using whole exome sequencing studies, five patients exhibited potential genetic predispositions and four patients had relevant germline mutations. The findings of this study provide valuable information for updated risk adapted treatment and personalized care of childhood MBs in our cohort series and in Taiwan.

## 1. Introduction

Medulloblastoma (MB) was classified into four core molecular subgroups (WNT, SHH, Group 3, and Group 4) by consensus in a conference in Boston, MA, USA [1] and was further categorized into 12 subtypes in 2017 [2]. Cavalli et al. further divided SHH tumors into four subtypes named SHH α, β, γ, and δ [2]. Among these four molecular subtypes, SHH α and β are subtypes of infants. SHH γ primarily affects non-infant children. Classification of the four main molecular subgroups was based on transcriptional profiling studies of MB cohorts. These subgroups were distinct in their demographics, clinical features (histology, metastasis, prognosis), genetics, and gene expression [1]. WNT subgroup is characterized by its peak incidence in late childhood, rare metastasis, very good prognosis, and activation of the WNT pathway. Patients may have *APC* germline mutation and predispose to Turcot syndrome [1,3,4]. SHH subgroup is characterized by activation of the SHH pathway and occur more frequent in infants (0–3) years and adults (>16 years). SHH MB patients in particular have high level expression of *GLI2* and *MYCN*. Non-metastasis *MYCN* amplification in SHH tumor is a marker of poor prognosis [5,6]. The associated germline of *TP53* and *PTCH1/SUFU* mutation predisposes to Li-Fraumeni syndrome and Gorlin syndrome, respectively [1,4,6]. Group 3 occur in infant and children. This subgroup is characterized having high incidence of LCA histology, high metastasis, *MYC* amplification, and poor outcome. However, patients with non-metastatic *MYC* amplified Group 3 tumor is still not considered high-risk [1,5,6]. Group 4 occurs with peak incidence at late childhood. It is characterized by frequent metastasis and moderate prognosis. Chromosome 11 loss is a prognostic marker of low-risk [6,7]. After the adoption of genetically defined subgroups of MB in the 2016 World Health Organization (WHO) classification [8], we initiated a project to recruit molecular diagnosis of childhood MBs for updated risk-adapted therapy in Taiwan. A cohort series of 52 cases of childhood MB with frozen tumor tissues were collected. Transcriptome and DNA methylation data were generated. With reference to the reports of Taylor et al. [1] and Cavalli et al. [2], we classified childhood MB into four main subgroups (WNT, SHH, Group 3 and Group 4) and SHH tumors were further designated to three subtypes (SHH α, β, γ). Furthermore, we analyzed the molecular-clinical correlation of this group of patients. Our purposes were to figure out the subgroup-specific clinical features, significant clinical and molecular prognostic markers, and survivals in our cohort series.

MB is the most common malignant brain tumor in children [9]. Before after the era of molecular classification, subtotal resection (STR) and presence of metastasis are the most significant clinical prognostic factors [10,11,12]. To evaluate the prognostic significance and potential genetic backgrounds of STR and tumor metastatic, we defined three molecular subgroups-based clinical risk stratification subgroups for survival and comparative gene expression analysis. On the basis of molecular classification and current survival rate in childhood MBs, a molecular and outcome-based risk stratification scheme was proposed as a future risk stratification scheme for non-infant children in the consensus conference at Heidelberg in 2015 [6]. Considering that both infant and non-infant childhood MBs share similar prognostic factors, an adjusted Heidelberg risk stratification scheme was defined for survival analysis. The purpose was to appraise its applicability in our cohort series as a new risk stratification for adapted therapy in infant and non-infant children.

Patients with childhood MB and other pediatric cancers may associated with genetic predisposition syndromes [1,3,4,6]. Waszak et al. [4] identified six genes with damaging germline mutations and genetic predisposition in MB. The prevalence was highest in SHH MBs (20%, 20/141). In our cohort series, potential genetic predispositions were observed in five patients with SHH tumors through relevant clinical findings. We were interested in identify somatic driver mutations and germline mutations for genetic predisposition in our cohort series as reference for genetic counselling and personalized care. For WES studies of somatic and germline mutations, we focused on 10 SHH patients with available samples of tumor and blood because of limited funding for sequencing in this project.

In 2015, we reviewed our hospital cohort series of 152 childhood MBs [13]. We observed that both STR and metastasis disease had prognostic significance in children aged >3–18 years. The 5-year overall survival (OS) of non-metastatic high-risk disease (residue tumor >1.5 cm^2^) disease and metastatic high-risk disease in children aged >3–18 years was 66.2% and 39.2%, respectively [13]. Effective treatment of high-risk childhood MBs was an unmet need in our patients. Both metastasis disease and tumor recurrence are high lethal and resistant disease in childhood MB [13,14]. The major sites of MB recurrence are leptomeningeal dissemination to brain and spine. We further performed comparative metastasis-associated genes/pathways in tumor with/without metastasis disease and in tumors with/with recurrence in our cohort series. The purpose was to figure out genes/pathways relating to metastasis for potential prognostic risk stratification and therapeutic targeting. On the other hand, DNA damage response (DDR) coordinate DNA repair and the activation of cell cycle checkpoints. Mutations in pathways of DDR [15,16] and other specific gene mutations may cause MB formation [3,4,17]. Moreover, therapeutic targets may be deciphered from the identified mutations [15]. Pugh et al. [18] and Robbins et al. [19] uncovered subgroup-specific somatic mutations and the affected pathways of MB through exome sequencing. We attempted to explore potential somatic and germline mutations relating to therapeutic targets. In 2018, Coudray at al. [20] reported using RNA sequencing (RNA-Seq) to detect somatic mutations in cancer genomes. Using RNA-Seq data analysis of patients with cancer, Wolff et al. [21] detected somatic variants in a panel of clinically relevant genes. Piskol et al identifed genomic variants from RNA-Seq data [22]. Referencing the reports of Coudray, Woff, and Piskol [20,21,22], we decided to apply RNA-Seq data analysis for variant calling in a selected list panel of 51 genes in our cohort series of 52 childhood MBs. This list consisted of genes related to DDR pathways [15,16], MB genesis pathways [17], and genetic predisposition for MBs and pediatric cancers [3,4]. The selection of these genes was based on our observation of five patients presenting clinical findings of potential genetic predisposition. The potentially relevant driver mutations were *TP53*, *MSH6*, *PTEN*, *PTCH1*, and *TERT* promotor. Zhang et al. [3] in 2015 reported 21 genes associated with autosomal dominant cancer-predisposition syndrome in pediatric cancers. Wazak et al. [4] in 2018 identified six genes as MB predisposition genes. Merging these three observations, 26 genes with relevance to genetic predisposition in MBs and pediatric cancers were selected as part of the 51-selected genes. Within these 26-selected genes, two genes (*BRCA1*, *BRCA2*) and four genes (*MSH6*, *PMS2*, *MLH1*, *MSH2*) were genes in homologous recombination and DNA mismatch repair, respectively. In 2012 Pugh et al. [18] uncovered subgroup specific somatic mutations and affected pathway of MB that included histone methyl transferases, chromatin remodeling, WNT pathway, and SHH pathway. The remaining of the 51-selected genes were chosen from pathways of DDR, cell cycle checkpoint, and subgroup specific pathway of MB genesis such as WNT and SHH.

The results may unravel our understanding of the molecular biology of childhood MBs in our cohort series. Moreover, this study can also help us move in the direction of providing adaptive, precise, and personalized therapy of childhood MBs in Taiwan.

## 2. Results

### 2.1. Patient Cohort

In this cohort series of 52 MBs in children, the median age at diagnosis was 7.2 years (range, 3.8 months to 18.2 years). Eight children (15.4%) were ≤3 years, and 44 (84.6%) children were >3 years (>3–18.2 years). The male to female ratio was 1.2/1 (28/24). In total 13 (25%) tumors presented metastasis at diagnosis. Near-total to total resection (NTR-TR) and STR were achieved in 29 (55.8%) and 23 (44.2%), respectively. Histological variants were classified as classic MB (n = 27, 51.9%), desmoplastic/nodular MB (n = 8, 15.4%), MB with extensive nodularity (n = 1, 1.9%), large cell/anaplastic (LCA) MB (n = 15, 28.9%), and MB with melanotic myogenic differentiation (n = 1, 1.9%).

Post-resection treatment strategies included chemotherapy alone (n = 3, 5.8%), radiotherapy alone (n = 4, 7.7%), and radiation plus chemotherapy (n = 45, 86.5%). In a median follow-up period of 5.1 years (range, 0.3 to 26.3 years), tumor recurrence occurred in 21 (40.4%) cases. The median time from diagnosis to first recurrence was 1.5 years (range, 0.2 to 19.5 years). The 5-year overall survival (OS) and relapse free survival (RFS) of the whole cohort series was 74% and 60.2% respectively (Figure 1a). Clinical data are summarized in Table 1.

### 2.2. Molecular-Classification and Molecular-Clinical Correlation

At the Medulloblastoma Down Under 2013 meeting [23], a consensus was reached that molecular subgrouping should be based on two independent validated analytical methods performed in accredited laboratories. Gene expression profiling and DNA methylation microarray are the current gold standard for MB subgrouping. In this study, we matched RNA-Seq and DNA methylation analysis results to classify 52 pediatric MBs into four core subgroups and three SHH pediatric subtypes. The molecular classification of our cohort series was consistent with the proposed molecular classification algorithm of MB.

Four core subgroups were identified as: WNT (n = 7, 13.5%), SHH (n = 17, 32.7%), Group 3 (n = 15, 28.8%), and Group 4 (n = 13, 25%). The SHH subgroup could be further subtyped as: SHH α (n = 4), SHH β (n = 7), and SHH γ (n = 6) (Figure 2a–e). Heatmap of expression profiling and the top and bottom 20 most differentially expressed genes in the three SHH subtypes was highlighted (Figure 2d and Appendix A). In the heatmap, common genes were found across the three SHH subtypes. The OS and RFS of the four core molecular subgroups showed no statistical significance with p value of 0.28 and 0.31, respectively (Figure 1b,c). However, the trend of OS in each subgroup matched with that reported by Shih et al. [7]. The molecular-clinical correlation was summarized in Table 1. The median age at diagnosis of molecular subtypes varied from 1.7 years in SHH γ to 10.1 years in Group 4 patients. The subgroup distribution of tumor resection, metastatic status at diagnosis, histological variants, treatment strategies, and tumor recurrence are presented in Appendix A. The STR rate ranged from 14.3% in SHH β to 61.5% in Group 4, and 75% in SHH α tumors; Group 4 and SHH α tumors had high STR rates (Appendix A). Among 13 cases with metastasis at diagnosis, the subgroup distributions were n = 0/7(0%) in WNT, n = 3/17 (17.6%) in SHH, n = 6/15 (40%) in Group 3, and n = 4/14 (30.8%) in Group 4 MBs (Appendix A). Third ventricular infundibular recess metastasis was observed in SHH and Group 4 tumors. For the 10 cases having metastasis at diagnosis, as determined by RNA-Seq data generated from primary tumors, the subgroup distributions were: one in SHH α, two in SHH β, four in Group 3, and three in Group 4. In each molecular subgroup of non-WNT tumors, comparing the three molecular-based clinical risk stratification subgroups, we observed subgroup-specific gene expression profiles and transition of gene expression profiles in metastatic high-risk tumors and non-metastatic high-risk (STR) tumors, respectively (Figure 2f–i). The results revealed that genetic profile might exist in both metastasis and STR.

Furthermore, we performed comparative gene expression profiles and gene set enrichment analysis of the top 20 metastasis-associated genes in different subgroups between tumors with/without metastasis (Appendix A) and between tumors with/without recurrence (Appendix A). Subgroup-specific upregulations of metastasis-associated genes and affected pathways were observed in in tumors with metastasis and also in tumors with recurrence. Clinically, both tumor with metastasis or recurrence presented treatment resistance and high lethality [13,14], we overlapped the top 20 upregulated metastasis-associated genes in tumor with metastasis and tumor with recurrence across difference subgroups (Appendix A). Metastasis-associated genes were identified in SHH α (*FABP1, SAA3P*), Group 3 (*BRD7, CYCS*), and Group 4 tumors (*EIF1AX, FOXO4, LRP6, MIA3, MYO5A, PRKAA1, YES1*). Subgroup-specific metastasis-associated genes and affected pathways may contribute to prognostication and also the development of targeted therapy for resistant metastatic disease and recurrent tumor. Subsequent analysis and investigation in large cohort is necessary.

For the metastatic tumor at diagnosis, our findings accord with the difference of cellular origin in MB subgroups [24,25] and also the report of Wu et al. [26] that MB metastatic tumor is genetically distinct from the primary tumor. The heterogeneity of gene expression among molecular subgroups, third ventricular infundibulum recess metastasis in both SHH and Group 4 MB observed in our cohort series, better survival of Group 4 than Group 3 tumors observed in the study of Shih et al. [7] and our cohort series, and the subgroups-specific expression profile of metastatic-associated genes/pathways in tumor metastasis or tumor recurrence may also imply subgroup-specific cell of origin and proportion of undifferentiated and differentiated neuronal-like populations [25].

The distribution of histological variants showed high proportion of desmoplastic tumors in SHH α and SHH γ MBs. Classic type was predominant in WNT and SHH β tumors. In Group 3 and Group 4 tumors, classic and LCA MBs were distributed equally (Appendix A). For postresection treatment, the three children with SHH γ MB received conventional chemotherapy alone, and all of them survived with a median follow-up period of 5.5 years (Appendix A).

In the whole cohort series, 21 (40.4%) tumors across molecular subgroups recurred in a median follow-up time of 5.1 years. The percentage of recurrence was low in WNT MB (n = 1, 14.3%) as compared with that in Group 4 (n = 4, 30.8%), SHH γ (n = 3, 50%), SHH β (n = 4, 57.1%), and Group 3 (n = 8, 53.3%) tumors (Appendix A). LCA MBs were found in seven (46.7%) Group 3 tumors and in six (46.2%) Group 4 tumors. Six of seven (85.7%) LCA tumors in Group 3 were associated with tumor recurrence compared with two of seven (33.3%) recurrences in Group 4 tumors. The result suggested that LCA Group 3 tumor may be very high-risk tumor.

### 2.3. Molecular Subgroup-Based Clinical Risk Stratifications and Survivals

Given the molecular subgroup-based clinical risk stratifications defined in this cohort study, the MB distribution was as follows: 25 cases in the non-metastasis average risk (Non-met AR) group, 14 cases in the non-metastasis high-risk (Non-met HR) group, and 13 cases in the metastasis high-risk (Met HR) group. The OS and RFS in these three groups of patients showed statistical significance with p value of 0.0047 and 0.0015, respectively (Figure 1d,e). The results showed that STR of M0-1 non-WNT tumors presented prognostic significance. These findings matched with our previous reports concerning a cohort series of 152 childhood MBs before molecular diagnosis [13].

### 2.4. Distribution of Selected CNVs and Chromosomal Aberrations across MB Subgroups

The subgroup distributions of selected molecular markers that we applied in the adjusted Heidelberg risk stratification scheme were shown in Figure 3 and Appendix A. In total, 50 MBs in our cohort series had data from DNA methylation array; the percentages of *MYCN* and *MYC* amplification in SHH and Group 3 of MBs were 19% (3/16) and 14% (2/14), respectively. Chromosome 11p loss in Group 3 and Group 4 tumors was 14% (2/14) and 23% (3/13), respectively. Iso17q was observed in 0% (0/14) of Group 3 and 23% (3/13) of Group 4 tumors. The prevalence of *TP53* mutation was 1.9% (1/52) that distributed in SHH subgroup with mutation rate of 5.9% (1/17). According to the report of Zhukova et al. in 2013, the mutation rates of *TP53* were high in WNT (16%) and SHH (21%), but absent in group 3 (0%) and rare in Group 4 MB (0.8%) [27]. In this study, the *TP53* sequencing was performed on the entire coding sequence (exons 2 through 11) comparing to the identification by RNA-Seq analysis in our cohort series. In the report of Northcott et al., similar results were found with *TP53* mutation rate of 16% in WNT and 13% in SHH MBs respectively. Waszak et al. [4] identified six genes that included *TP53* as MB predisposing genes. This study found *TP53* germline mutations in 14 (1%) of all 1022 patients with MB and were only identified in SHH MBs. In our cohort study, the only one *TP53* mutation distributed in SHH subgroup and presented as somatic and germline mutations. The prevalence of germline mutation of *TP53* in our series was 1 (1.9%) of 52 patients.

### 2.5. Survival Analysis of the Adjusted Heidelberg Scheme

Regarding the adjusted Heidelberg risk stratification scheme (refer to Table 2 in the materials and methods section), the distribution of patients in different risk subgroups was n = 7 in low-risk group, n = 27 in standard-risk group, n = 8 in high-risk group, and n = 7 in very high-risk group. Survival analysis showed statistical significance in both OS and RFS with p value of 0.0049 and 0.032, respectively. The 5-year OS in each risk subgroups corresponding to the defined survival rates in the scheme. In OS analysis, statistical significance was observed only between low-risk and high-risk group. In RFS analysis, statistical significances were observed between low-risk and high-risk groups (*p* value = 0.0008), between standard-risk and high-risk groups (*p* value = 0.0034), and between high-risk and very high-risk groups (*p* value = 0.0011). The 5-year survival of the high-risk group dropped to less than 50% and closed to the survival curve of the very high-risk group (Figure 1f,g).

The Heidelberg risk stratification scheme was proposed in 2015 as a future risk stratification scheme for non-infant childhood MBs [6]. Considering that both infant and non-infant childhood MBs share similar prognostic factors, we evaluate the clinical applicability of the adjusted Heidelberg scheme in childhood MBs of all ages in our cohort series. In survival analysis, the 5-year OS corresponding to the survival rate defined in different risk groups. The results supported the applicability of this adjusted scheme (Table 1, Figure 1f,g) in our cohort series. The 5-year RFS analysis of the high-risk group was less than 50% and closed to that of the very high-risk group. Group 3 LCA MB presented a very high-risk of recurrence in six of seven tumors of our cohort series and may be considered to be a very high-risk tumor in the Heidelberg scheme. On the base of the establishment of molecular diagnosis and the adjusted Heidelberg scheme, new nationwide risk-adjusted treatment protocols for childhood MBs have been established in Taiwan with the support of Taiwan Pediatric Oncology Group (TPOG).

### 2.6. Somatic Mutations and Their Subgroup Distribution

In a selected panel of 51 genes, somatic mutations were detected in 46 of them by using RNA-Seq data. The mutations prevalence in these 46 genes ranged from 1.9% to 30.8% in *KMC2D* (Table 3). Many of these gene showed subgroup-specific distribution. The affected signature pathways of these genes were the following: chromatin remodeling, DNA damage checkpoint, DNA repair, homologous recombination, DNA mismatch repair (MMR), histone methyltransferase, WNT pathway, SHH pathway, and others that included *TP53, PTEN,* and *NOTCH1/2*. Significant somatic mutations in pathways of DDR were *PARP1, ATM/ATR, BRCA1/2, MSH6, PMS2, KMT2C/D*, and *TP53*. Driver mutations of WNT-activated (*CTNNB1,* and *DDX3X*) and SHH-activated (*PTCH1, SMO,* and *SUFU*) MBs were identified. Moreover, other driver mutations included *APC, PTEN,* and *NOTCH1/2* were also detected. *TERT* promoter mutation could not be detected through RNA-Seq data. The identified somatic mutations, affected pathways, and subgroup distribution are shown in Figure 4a. In gene set enrichment analysis and annotation based on Gene Ontology (GO) and the Kyoto Encyclopedia of Genes and Genomes (KEGG) pathway database, the affected signature pathways showed significant correlation with WNT and SHH MBs (Figure 4b). Mutation frequency were calculated per mega base pair in our cohort series of 52 cases (Figure 4c). SHH tumors exhibited the highest mutation frequency (9.8/Mb), whereas Group 4 tumors exhibited the lowest frequency. Furthermore, SHH γ exhibited the highest mutation frequency (11.7/Mb) in SHH tumors (Figure 4d).

Detection of somatic mutation in cancers by using RNA-Seq data has the benefit of having no additional cost after the initial sequencing [18,19]. Using limited resource, we applied RNA-Seq variant calling to detect somatic mutations in our cohort series. With reference to the report of Woff et al. [21], within a selected panel of 51 genes, we identified 46 somatic mutations (Figure 4a, Table 4). Some of these somatic mutation and affected pathways matched with the observations of Pugh et al. [18], with somatic mutations identified through WES. For potential clinical applications, these 46 identified somatic mutations were divided into two categories: (1) Mutations in pathways of DDR for the potential application in therapeutic targeting. (2) Specific driver mutations in MB genesis and genes associated with a genetic predisposition of MB and pediatric cancer for potential application in personalized care and genetic counseling. We further analyzed the correlation of top 20 upregulated and downregulated genes in four core subgroups and tumor mutation burden (Appendix A). The correlation coefficient and p value were summarized in Appendix A.

Interestingly, we found that in the SHH group, most of the top 20 upregulated genes had positive associations with the mutation numbers, while all the top 20 downregulated genes had negative associations. On the contrary, in Group 4, all the top 20 upregulated genes had negative associations with the mutation numbers, while all the top 20 downregulated genes had positive associations. This reflects the distinct molecular mechanism of the two subtypes, offering novel angles of future investigations. To our knowledge, this phenomenon has not been reported before.

### 2.7. Potential Association of Genetic Predisposition in Patients

In this cohort series of 52 MBs, 17 were SHH MBs. According to relevant medical records, five patients harbored SHH MBs and their clinical findings might be linked with a potential genetic predisposition. Relevant somatic mutations identified based on RNA-Seq calling in four of these five patients involved *TP53, MSH6, PTEN*, and *PTCH1*. The suspicious *TERT* promoter mutation in the remaining one patient was not detected. For validation, whole exome sequencing (WES) of blood and tumor tissue was used in 10 patients with SHH MBs in the cohort series. This study included the five patients with potential genetic predispositions. Germline mutations were explored through WES of blood samples in these patients. *TP53*, biallelic *MSH6*, and *PTEN* germline mutations were identified in three of these five patients with potential genetic predisposition (Appendix A). The *SUFU* germline mutation was identified in one the remaining five SHH MBs in patients without relevant clinical findings of potential genetic predisposition (Appendix A). The relevant somatic mutations, germline mutations, clinical features, potential genetic predisposition, and mutational frequency of these six patients with potential genetic predisposition were summarized in Table 4. These six patients presented one of the following clinical and molecular features: (1) A high-risk SHH α M3 tumor with early recurrence (Case 1, *TP53* germline mutation; Appendix A). (2) Constitutional mismatch repair deficiency (CMMR-D) with nonsynchronous diagnosis of three malignancies including SHH α MB, anaplastic astrocytoma, T-lymphoblastic leukemia, and blue nevi (Case 2, *MSH6* biallelic germline mutation; Appendix A). (3) Suspicious *PTEN* tumor hamartoma syndrome (PTHS) with synchronous diagnosis of recurrent SHH β MB and multiple adenomatous hyperplasias/nodular goiter (Case 3, *PTEN* germline mutation; Appendix A). (4) Synchronous diagnosis of two malignancies-parotid rhabdomyosarcoma and SHH γ MB (Case 4, *PTCH1* somatic mutation). (5) Nonsynchronous occurrence of SHH β MB and buccal myxofibrosarcoma (Case 5, no relevant somatic mutation identified, *TERT* promoter mutation was considered, unvalidated). (6) Germline mutations of *SUFU* and *PALB2* identified in a 2.2 years old boy with SHH γ MB but no other clinical trait of Gorlin syndrome was recorded (Case 6, Appendix A). The loci of somatic and germline mutations of 10 patients with SHH MB (including the five cases with relevant clinical findings) were shown in Appendix A.

## 3. Discussion

### 3.1. Potential Therapeutic Targeting from Mutations of the DDR

Exogenous and endogenous DNA damage contribute to germline or somatic DNA repair deficiencies in cancer [28]. The identification of somatic mutations involving DDR may help in targeted therapy with a genomic approach [29,30]. We identified somatic mutations in DNA repair pathways of single-strand breaks (SSBs) and double-strand breaks (DSBs). In SSBs, the detected mutations included PARP in base resection repair (BER) and *MSH6*/*PMS2* in MMR. In DSBs, the identified mutations included *BRCA1/2* and *ATM/ATR*. In patients with dysfunction of each DNA repair pathway, therapeutic targets for synthetic lethality may be exploited to inhibit the compensatory DDR pathway. The *RAD52* inhibitor may have extended synthetic lethality in *BRCA1/2* mutation/deficiency tumors [31]. BRCA-mutated tumors may be treated by targeting *ATR/CHK1* axis with the PARP inhibitor [32]. Evidence has indicated the synthetic lethality of using the DNA-PCK inhibitor [33], Pol β inhibitor [34], and PARP inhibitor [35] from cell line or preclinical studies in tumors with MMR deficiency and somatic mutations.

Upregulated DDR pathways confer to resistance of DNA damage therapy and DDR inhibitor may overcome resistance [15]. DNA repair/cell cycle checkpoint genes including *PARP1, BRCA1, ATM* and *TP53* may also be associated with tumor metastasis [36]. In our cohort series, the prevalence of somatic mutations of these four genes was *PARP1* (9.6%), *BRCA1* (5.8%), *ATM* (19.2%), and *TP53* (1.9%) respectively. *TP53* was the most significant risk factor in SHH MBs. On the other hand, incidence of metastasis of MB by subgroup varies from 17.9% in WNT, 19.1% in SHH, 46.5% in group3, and 29.7% in group 4 tumors in the report of Northcott et al. in 2011 [37]. Metastasis is the most significant prognostic factor of non-WNT childhood MBs. *PARP1* inhibitor may be a promising treatment for some MB with DDR dysregulation or for metastatic or resistant MBs. Further preclinical study in metastatic MB cell lines and mice model is required.

### 3.2. Finding Clues to Genetic Predisposition

Waszak SM et al. [4] identified six genes with damaging germline mutations and genetic predisposition in MB. The prevalence was highest in SHH MBs (20%, 20/141). These six genes were *APC, BRCA2, PALB2, PTCH1, SUFU*, and *TP53*. Except for a heterozygous germline mutation in the *MSH6* of a patient, no biallelic germline mutation of MMR genes were identified. However, among the 21 germline mutations with a genetic predisposition in pediatric cancer, *PMS2, MSH2,* and *MSH6* were identified [8]. Individuals heterozygous mutations of *BRCA2* have increased risk of inherited breast and ovarian cancer [38]. Germline heterozygous mutation of *BRCA2* and *PALB2* may be associated with an increased risk of childhood MB [4,39]. Recurrent mutations of *PTEN* were detected in SHH MBs [17]. The high frequency of *TERT* promoter mutation occurred in SHH MBs of adult patients [40]. Genetic syndromes with germline mutations associated with MBs [41,42,43] included Gorlin syndrome (heterozygous germline pathogenic variant in *PTCH1* or *SUFU*) [44,45], CMMR-D syndrome (biallelic deleterious germline mutations in MMR genes (*MLH2, MSH2, MSH6,* and *PMS2*)) [46], Li-Fraumeni syndrome (germline mutation of *TP53*) [47,48], familial adenomatous polyposis syndrome (germline *APC* muttaion) [4,49], and Cowden syndrome (germline mutation of *PTEN*) [41,50]. In our cohort series, we identified somatic mutations in *BRCA2, PALB2, MSH6, PMS2, PTCH1, SUFU, TP53, APC,* and *PTEN* from RNA-Seq data analysis. *TERT* promotor mutation was not detected by RNA-Seq data.

According to relevant clinical findings in our cohort series, patients with potential genetic predisposition were observed in five cases that clustered only in SHH tumors (Table 4). *TP53* mutations was the most significant risk factor in SHH MBs [27]. Poor survival of non-subgrouped childhood MB with somatic mutations of *TP53* was reported [51]. Case 1 had a germline mutation of *TP53*. The clinical course of this patient matched with the reported studies (Appendix A). CMMR-D is a rare syndrome causing by biallelic deleterious germline mutation in MMR genes. Carriers of biallelic *MSH6*/*PMS2* mutations have higher rates of brain tumors and Lynch syndrome associated tumors [46]. A child having biallelic *MSH6* mutations was reported to have MB at the age of 7, acute myelocytic leukemia at 10 years, and coloric polyps/carcinoma at 13 years [52]. We identified the *MSH6* somatic mutation and biallelic germline mutation in case 2. The sequantial development of three malignancies corresponded to the case report of CMMR-D (Appendix A). In the *PTEN* and Trp53 double knockout mice model, fully penetrated MB developed from perivascular progenitor niche in postnatal cerebellum [53]. Moreover, *PTEN* mutations were identifed in MBs [54,55]. Cowden syndrome is an autosoma dominant cancer predisposition syndrome caused by the germline mutation of *PTEN* [50]. In case 3, the child presented *PTEN* germline mutation and *TP53* deletion. He exihibited the synchronous diagnosis of recurrent SHH β MB and multiple adenomatous hyperplasia/nodular goiter. However, the relevant clinical findings did not fullfil the diagnostic criteria of Cowden syndrome [50]. The co-existence of *PTEN* germline mutations and *TP53* deletion may link with the genesis of MB and genetic predisposition (Appendix A). Gorlin syndrome (nevoid basal cell carcinoma syndrome, NBCCS) is caused by heterozygous *PTCH1* or *SUFU* germline mutations [45,56]. Approximately 1–2% of MBs are attributable to this syndrome [57] and both fetal rhabdomyosarcomas and rhabdomyosarcomas were observed in NBCCS patients [58]. Case 4 had a *PTCH1* somatic mutation but no germline mutation. She presented two non-synchronous malignancies, namely parotid rhabdomyosarcoma and SHH γ MB. The potential assocition with a genetic predispition should still be considered. In case 5, the patient had heterochronous occurrence of SHH β MB and buccal myxofibrosarcoma. Somatic mutations of *PARP1*, *KMD2C*/*D,* and *NOTCH2* were identified. No single mutation was linked to the development of the two tumors. In a report by Kellela et al. [59], the frequency of *TERT* promoter mutations in MB was 19/91 (20.8%) and in myxofibrosarcoma was 1/10 (10%). *TERT* promoter mutation might be the driver gene of the two tumors in this patient, but it could not be detected in RNA-Seq data. Among the other five cases of SHH MB that had no relevant clinical findings of potential genetic predisposition, *SUFU* germline mutations were identified in a case of SHH γ MB (Appendix A, Table 4, Case 6). This case may be a rare germline *SUFU* mutation and a candidate for genetic counseling and surveillance for Gorlin syndrome.

## 4. Materials and Methods

### 4.1. Patient Cohort

This was a cohort study of Taipei Veterans General Hospital and Taipei Medical University Hospital on MBs in children aged <20 years at diagnosis. High quality frozen tumor tissue samples from 52 patients were retrieved in between 1989–2017. The frozen tumor tissue consisted of 48 cases of primary tumors and another 4 cases of first recurrent tumors for molecular subgroup classification. We did not have frozen tissue collected from the primary tumors in these four patients. The recruitment of recurrent tumors for subgroup classification was based on the report [60] that MB did not change its molecular classification at recurrence. To avoid difference of gene expression between the primary and recurrent tumors, analysis of subgroup-specific gene expression in tumors with metastasis at diagnosis was performed in 10 patients with RNA-Seq data from primary tumors. All subjects gave written informed consent in accordance with the Declaration of Helsinki. The sample were fully encoded and used under a protocol approved by the Institutional Review Board of Human Subjects Research Ethics Committee of the Taipei Medical University Hospital and Chang Gung Memorial Hospital, Taiwan. The IRB approval number is 201701441A3.

### 4.2. Retrieval of Clinical Data

The clinical data were retrieved from medical records. Histological diagnosis was classified according to the WHO 2007 and WHO 2016 classifications [8,61] and reviewed by three neuropathologists (D.M.T.H., S.C.L., and W.C.H.). Key perioperative and follow-up tumor images were reviewed by two neuroradiologists (Feng-Chi Chang and Kevin Li-Chun Hsieh). Cerebrospinal fluid cytology for metastasis in the perioperative period was not a routine procedure in our cohort series. We defined the status of metastasis at diagnosis as M0-1 and M2-3 according to Chang’s operative staging system [62]. The status of metastasis at diagnosis was classified as M0 (no metastasis), M1 (presence of tumor cells in CSF), M2 (seeding in the intracranial subarachnoid space and the intracranial compartment), and M3 (seeding in the spinal subarachnoid space). The extent of tumor resections was classified as STR (residue tumor ≥1.5 cm^2^) and NTR-TR (residue tumor <1.5 cm^2^) by using postresection tumor images [63]. We did not perform further resection after primary STR. Post-resection treatment strategies were grouped as chemotherapy alone, radiation therapy alone, and radiotherapy with chemotherapy. The diagnosis date was defined as the date of first tumor resection. The diagnosis date of first recurrence was defined as the date of first recurrence found in longitudinal follow-up tumor images. Relevant clinical findings relating to genetic predispositions were defined as follows: (1) High malignancy and early recurrence for poor prognosis of somatic/germline *TP53* mutation in SHH MB. (2) Other synchronous or nonsynchronous malignancy. (3) Associated hamartomatous lesions in germline *PTEN* mutation.

### 4.3. Molecular Diagnosis

RNA-Seq and methylation profiling were performed for 52 and 50 cases, respectively, in the 52-case cohort series. RNA-Seq was run in two lanes of a Nextseq 500 sequencing instrument (Illumina) for multiplexed paired-end reads. The gene expression table were extracted using Kallisto [64] and tximport [65] package in R environment. For clustering, unsupervised clustering analysis was performed on the basis of 10,000 most differentially expressed genes using the consensus clustering default parameters through Rtsne, NMF [66], and ConsensusClusterPlus [67]. The result was validated using 22 subgroup-specific signature genes expression levels [68]. Downstream clustering and differential expression analysis were performed using DESeq2 [69]. Taylor’s laboratory in the Hospital for Sick Children, Toronto helped with counterpart clustering. EPIC methylation profiling and cluster analysis in 50 MBs in this cohort series were performed in Nada Jabado’s laboratory at the Research Institute of the McGill University Health Center. Raw data files were read and preprocessed using the capabilities of minfi [70] and ChAMP [71]. Unsupervised clustering analysis was performed on the basis of 10,000 most differentially expressed probes using consensus clustering and validated by using 11 subgroup-specific signature probes [72]. Subtype clustering of SHH α, β, and γ was performed based on the top 1% of the most differentially expressed common genes (n = 201) and probes list from a study [2]. Subgroup specific focal CNVs and chromosomes aberrations with prognostic significance were selected from reported large cohort studies and review articles [6,7,73] as molecular prognostic markers. CNVs were identified from methylation arrays by using conumee [2,74,75]. The log2 ratio of chromosomes or genes more than 0.2 was defined as gain and that of less than −0.2 was defined as loss. A panel of 51 clinically relevant mutations was selected for detecting somatic mutations in RNA-Seq genome of 52 MBs. These selected mutations were linked to DDR, MB genesis, a genetic predisposition for MB, and pediatric cancers predisposition syndromes [3,4,15,16,17,53]. Raw RNA-Seq results were aligned using HISAT2 [76], and this was followed by variant calling using the HaplotypeCaller tool in GATK. Variants were annotated using ANNOVAR [77] based on COSMIC database [78]. Candidate somatic driver mutations with potential genetic predisposition associations were assessed through correlating the relevant clinical findings of the patients. For detecting and validating germline mutations in SHH MBs, subsequent WES was performed in 10 patients of SHH MBs with available blood and tumor tissues. These 10 patients included those having relevant clinical findings of a potential genetic predisposition. Raw reads from WES were aligned to the human genome build hg38 using the Burrows-Wheeler Aligner [79] in patients with a potential genetic predisposition. Somatic variant calling and filtering were performed using VarDict [80], and all variants in IGV with alignment level were visualized [81]. Mutation gene set enrichment analysis was performed using goseq [82] and MsigDB gene set collections, namely Gene Ontology (GO) and the Kyoto Encyclopedia of Genes and Genomes (KEGG). Mutation frequency were calculated per mega base pair with RNA-Seq data. The signature between variables was determined using the Kruskal-Wallis test.

### 4.4. Molecular Subgroup-Based Clinical Risk Stratification

In our previous study of 152 childhood MBs, both STR and metastasis (M2-3) at diagnosis were poor prognostic factors in childhood MBs in those aged >3 years [13]. Nonmetastatic WNT MB is classified as low-risk tumor with a >90% survival rate [11,12]. We attempted to integrate WNT MB, non-WNT MB, EOR (NTR-TR and STR), and metastasis status at diagnosis (M0-1, M2-3) to define molecular subgroup-based clinical risk stratification in our cohort series. This risk stratification was defined as follows: (1) Non-met AR in WNT (M0-1) or non-WNT (M0-1, NTR-TR) tumors; (2) Non-met HR in non-WNT (M0-1, STR) tumors; and (3) Met HR in non-WNT M2-3 tumors. OS and RFS were analyzed in these three risk groups for statistical significance. Furthermore, their subgroup-specific gene expressions were analyzed for the genetic backgrounds of metastasis at diagnosis and the EOR.

### 4.5. The Adjusted Heidelberg Risk Stratification Scheme

To adopt the Heidelberg scheme, we made minor adjustments (Table 2). The adjustments included incorporating the nonmetastatic (M0-1) tumor instead of M0 tumor because we did not routinely study CSF cytology in the perioperative period. Nonmetastatic (M0-1) SHH tumor, Group 3 tumor without data of *MYCN* and *MYC* amplification, and Group 3 tumor with *MYC* amplification, respectively, were defined as undetermined-risk tumors (n = 1). LCA tumors in Group 3 (n = 7) and Group 4 (n = 6) were not defined as being undetermined-risk or high-risk tumors. Survival analysis of this adjusted scheme was performed to evaluate its applicability in all ages of childhood MBs.

### 4.6. Survival Analysis

OS and RFS analysis were based on the date of first tumor surgery (diagnosis date), date of first recurrence, date of last follow-up, and date of death. Survival analysis was performed using the Kaplan-Meier method and drawn using survminer package in the R environment. The differences in survivals were assessed using the log-rank test. The association between categorized variables was determined using ANOVA and Kruskal-Wallis test. A *p* value <0.05 was considered statistically significant. With reference to the adjusted Heidelberg risk stratification scheme, survival analysis was performed.

## 5. Conclusions

Molecular-clinical correlation aids in unraveling molecular-based characteristics of MB in children. In this cohort study, we established our laboratory investigation of molecular analysis and classification of pediatric MBs. After this achievement, we entered the era of molecular diagnosis, molecular and outcome-based risk stratification, and adapted treatment according to the adjusted Heidelberg risk stratification scheme. New risk-stratified treatment regimens for MBs in infant and non-infant children have been established since 2017 under the guidance of TPOG to promote nationwide application. Detecting somatic mutations and affected pathways from RNA-Seq data is useful for assessing targeted therapy without incurring additional cost after sequencing. Correlating somatic driver mutations and relevant clinical findings of a patient may help to find clues to identify potential genetic predisposition for genetic counseling and prolonged clinical surveillance of patient and families. Because of the high prevalence of genetic predisposition in SHH MBs, WES for somatic and germline mutation will be valuable for validating hereditary genetic predisposition syndrome of MB in children. The observed subgroup-specific gene expressions in metastatic tumor at diagnosis and in subtotal tumor resection requires validation in large cohort studies.

## Figures and Tables

**Figure 1 cancers-12-00653-f001:**
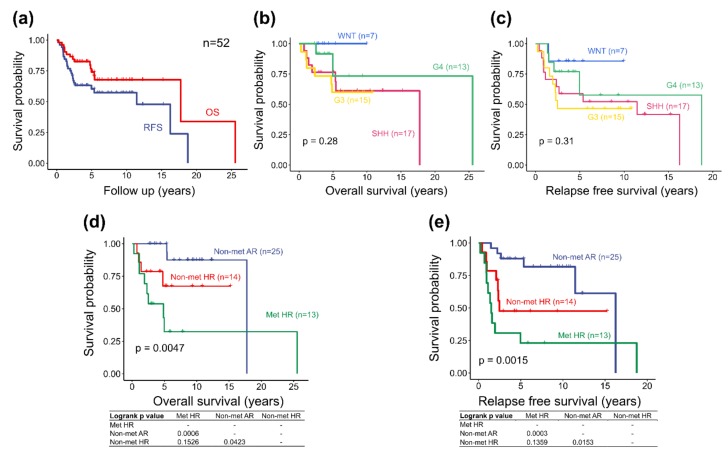
Outcome analysis of 52 childhood Medulloblastomas (MBs) in a cohort series in Taiwan. (**a**) Outcome analysis split based on overall survival (OS) (red) and relapse-free survival (RFS) (blue). (**b**,**c**) OS and RFS split into four core molecular subgroups: blue, WNT MB; green, Group 4 MB; red, SHH MB; yellow, Group 3 MB. (**b**) OS. (**c**) RFS). (**d**,**e**) OS and RFS split into three subgroups of molecular subgroup-based clinical risk stratification: blue, Non-met average risk (AR); red, Non-met high-risk (HR); and green, Met HR. (**f**,**g**) Survival analysis of the adjusted Heidelberg Scheme in childhood MBs of all ages split into blue, low-risk (LR); red, standard-risk (SR); green, high-risk (HR), and purple (very HR). (**f**) OS. (**g**) RFS.

**Figure 2 cancers-12-00653-f002:**
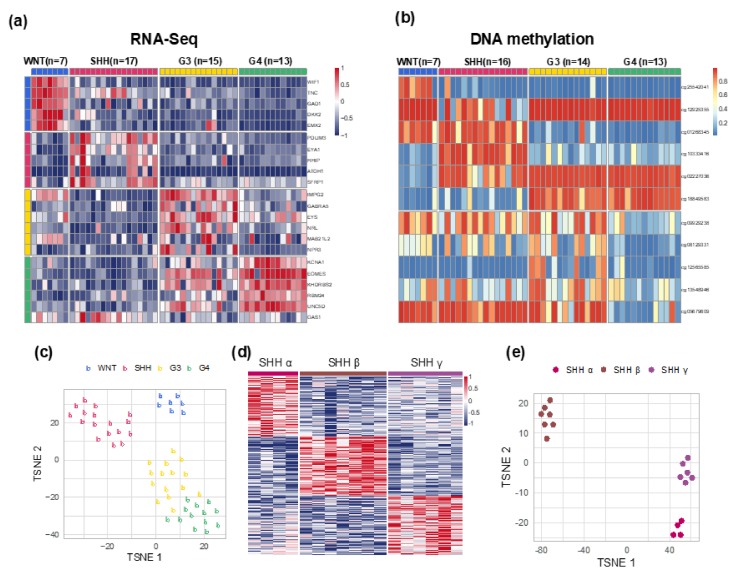
Molecular subgroups and the distribution of clinical parameters in the cohort series of 52 pediatric MBs in Taiwan. (**a**) Subgroup classification by using RNA-Seq. (**b**) Subgroup classification by using DNA-methylation array. (**c**) t-SNE dimensional distribution of the four core molecular subgroups. (**d**) Subtype classification of SHH subgroup. (**e**) t-SNE dimensional distribution of SHH subtypes. (**f**–**i**) MBs with metastasis found at diagnosis and RNA-Seq performed from primary tumor tissues. Comparing with Non-met AR tumors, subgroup-specific gene expression profiles in Met HR and transition of gene expression profile in Non-met HR tumors were seen across non-WNT molecular subgroups. (f) SHH α MB (Met HR, n = 1). (**g**) SHH β MB (Met HR, n = 2), (**h**) Group 3 (Met HR, n = 4). (**i**) Group 4 (Met HR, n = 3).

**Figure 3 cancers-12-00653-f003:**
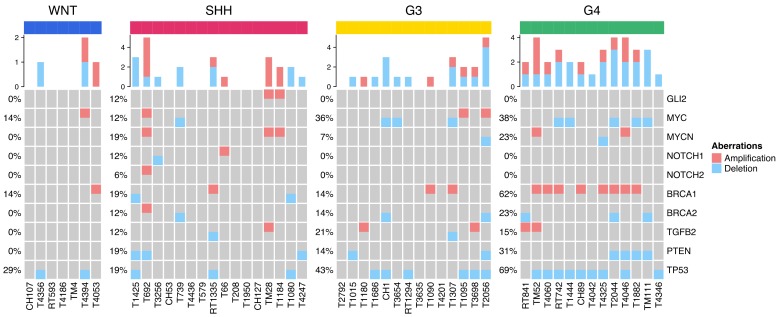
Subgroup distribution of gene CNVs of 50 MBs with DNA methylation analysis split into blue (loss) and red (gain).

**Figure 4 cancers-12-00653-f004:**
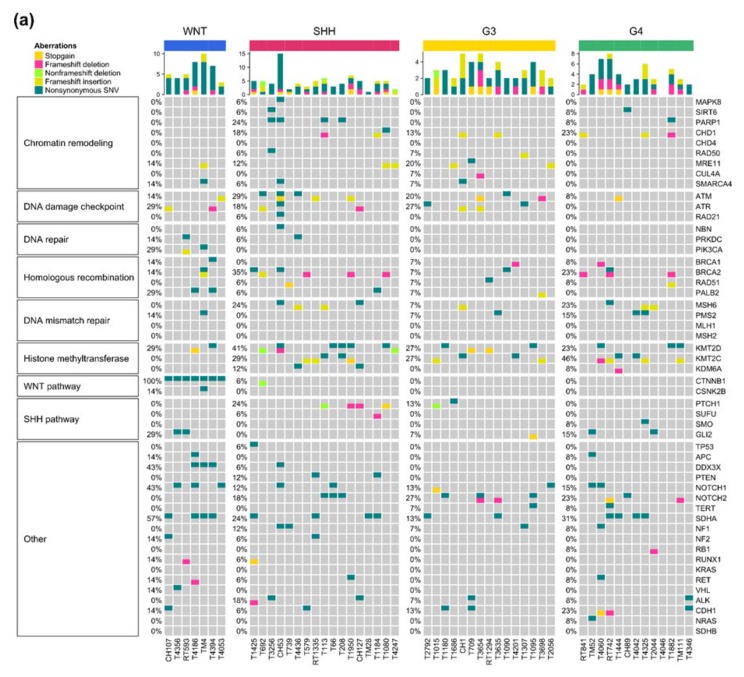
Calling of somatic mutations based on RNA-Seq in a selected list panel of 51 clinically relevant genes in the cohort series of 52 childhood MBs. (**a**) Prevalence and subgroup distributions of the somatic mutations. (**b**) Subgroup distribution of affected signature pathways. Significant signature pathways were identified based on gene set enrichment analysis and annotation was identified based on Gene Ontology (GO) and the Kyoto Encyclopedia of Genes and Genomes (KEGG) pathway database. Mutation frequency in the four core molecular subgroups (**c**) and SHH subtypes (**d**). SHH tumors carried the highest mutation frequency (9.8/Mb) (*p* = 0.31, Kruskal-Wallis test). Among SHH subtypes, SHH γ carry the highest mutation frequency (11.7/Mb) (*p* = 0.16, Kruskal-Wallis test).

**Table 1 cancers-12-00653-t001:** Demography, molecular subgroup, clinical data, and molecular-clinical correlation in our cohort series of 52 childhood Medulloblastomas (MBs) in Taiwan.

Molecular Subgroup Assignment						
N = 52	WNT n = 7 (13.5%)	SHH n = 17 (32.7%)			Group 3 n = 15 (28.8%)	Group 4 n = 13 (25%)
		SHH α, n = 4	SHH β, n = 7	SHH γ, n = 6		
**Frozen tissue for molecular profiling**						
Primary tumor	7	4	7	6	12	12
Recurrent tumor	0	0	0	0	3	1
**Age (median, range) at diagnosis (years)**						
7.2 (0.3–18.2)	8.5 (3.1–11.4)	4.0 (0.3–14.3)			6.0 (1.6–18.2)	10.1 (5.1–15.4)
		7.4 (2.8–12.6)	4.2 (3.7–14.3)	1.7 (0.3–5.8)		
≤3 y (n = 8, 15.4%)	0	1 (25%)	0	5 (83.3%)	2 (13.3%)	0
>3 y (n = 44, 84.6%)	7 (100%)	3 (75%)	7 (100%)	1 (16.7%)	13 (86.7%)	13 (100%)
**Sex**						
Male, n = 28 (53.8%)	1	2	4	4	9	8
Female, n = 24 (46.2%)	6	2	3	2	6	5
Male/female ratio (1.2/1)	0.2/1	1.4/1			1.5/1	1.6/1
		1/1	1.3/1	2/1		
**Metastasis stage at diagnosis (M0-1, M2-3), number of cases (percentage)**						
M0-1, n = 39 (75%)	7 (100%)	3 (75%)	5 (71.4%)	6 (100%)	9 (60%)	9 (69.2%)
M2-3, n = 13 (25%)	0	3 (17.6%)			6 (40%)	4 (30.8%)
		1 (25%)	2 (28.6%)	0		
**Extent of resection (NTR-TR, residue tumor < 1.5 cm2; STR, residue tumor ≥ 1.5 cm2), number of cases (percentage)**						
NTR-TR (n = 29, 55.8%)	5 (71.4%)	1 (25%)	6 (85.7%)	3 (50%)	9 (60%)	5 (38.5%)
STR (n = 23, 44.2%)	2 (28.6%)	3 (75%)	1 (14.3%)	3 (50%)	6 (40%)	8 (61.5%)
**Pathology variant, number of cases (percentage) and median age (years)**						
Classic, n = 27 (51.9%), 6.3	5 (71.4%),	1 (25%)	6 (85.7%)	2 (33.3%)	7 (46.7%)	6 (46.2%)
DNMB, n = 8 (15.4%), 2.8	1 (14.3%)	2 (50%)	1 (14.3%)	3 (50%)	0	1 (7.7%)
MBEN, n = 1 (1.9%), 1.0	0	0	0	1 (16.7%)	0	0
LCA, n = 15 (28.8%), 8.5	1 (14.3%)	1 (25%)	0	0	7 (46.7%)	6 (46.2%)
MMMB, n = 1 (1.9%), 4.3	0	0	0	0	1 (6.7%)	0
**Molecular subgroup-based clinical risk stratification *, number of cases**						
Non-met (M0-1) AR, n = 25	7	1	5	3	6	3
Non-met (M0-1) HR, n = 14	0	2	0	3	3	6
Met (M2-3) HR, n = 13	0	1	2	0	6	4
**Treatment strategy, number of cases (percentage)**						
CMT alone, n = 3 (5.8%)	0	0	0	3	0	0
RT alone, n = 4 (7.7%)	0	0	0	1	2	1
RT + CMT, n = 45 (86.5%)	7	4	7	2	13	12
**Recurrence, number of cases (percentage**)						
Recurrence, n = 21 (40.4%)	1 (14.3%)	8 (47.1%)			8 (53.3%)	4 (30.8%)
		1 (25%)	4 (57.1%)	3 (50%)		
**Time from diagnosis to the first recurrence (years)**						
Median time (range)	1.5	0.7	1.9 (0.9–16.3)	0.9 (0.4–2.3)	1.8 (0.2–2.5)	1.8 (1.4–19.5)
**Median follow-up time (range) (years)**						
5.2 (0.3–25.6)	3.7(2.7–10.0)	5.8 (1.1–15.3)	12.3 (1.9–17.7)	4.0 (0.8–10.5)	6.5 (0.3–10.9).	4.3 (1.7–25.6)
**Survivals of molecular subgroup (percentage)**						
5-year OS rate: 74.1%	100%	76.5%			60.0%	72.7%
5-year RFS rate: 60.4%	85.7%	58.8%			46.7%	61.4%

TR: Total resection, NTR: Near total resection, STR: Subtotal tumor resection, DNMB: Desmoplastic/nodular medulloblastoma, MBEN: Medulloblastoma with extensive nodularity, LCA: Large-cell/anaplastic, MMMB: Medulloblastoma with melanotic myogenic differentiation, Non-met: Nonmetastasis, Met: metastasis, M0-1: no metastasis to presence of tumor cells in CSF, M2: intracranial subarachnoid space or intracranial compartment metastasis, M3: intraspinal subarachnoid space metastasis, AR: average-risk, HR: high-risk, CMT: Chemotherapy, RT: Radiotherapy, OS: Overall survival, RFS: Relapse-free survival, * Refer to the materials and methods section.

**Table 2 cancers-12-00653-t002:** Adjusted Heidelberg risk stratification scheme for survival analysis at all ages of the 52 childhood MBs in a cohort series in Taiwan.

Subgroup/Survival Rate	Low-Risk (>90% Survival)	Standard-Risk (75–90% Survival)	High-Risk (50–75% Survival)	Very High-Risk (<50% Survival)	Undetermined
WNT	<16 y, Non-metastatic (M0-1)				Metastatic (M2-3)
SHH		All of the following: *TP53* WT (somatic), No *MYCN* amplification, Non-metastasis (M0-1)	One or both: *MYCN* amplification, Metastatic (M2-3)	*TP53* mutation (metastatic or non-metastatic)	No data of *MYCN* amplification
Group 3		All of the following: No *MYC* amplification, Non-metastatic (M0-1)		Metastatic (M2-3)	No data of *MYC* amplification, or Non-metastatic with *MYC* amplification
Group 4	All of the following: Non-metastatic (M0-1), Chr. 11 loss	All of the following: Non-metastatic (M0-1), No Chr. 11 loss	Metastatic (M2-3)		

Risk stratification was defined based on 5-year survival rate.

**Table 3 cancers-12-00653-t003:** Prevalence and subgroup distribution of the 46 somatic mutations identified based on RNA-Seq from a selected panel of 51 genes in a cohort series of 52 childhood MBs in Taiwan.

Pathway	Gene	WNT, n (%)	SHH, n (%)	G3, n (%)	G4, n (%)
Chromatin remodeling	*MAPK8* (1.9%)	0 (0%)	1 (1.9%)	0 (0%)	0 (0%)
	*SIRT6* (3.8%)	0 (0%)	1 (1.9%)	0 (0%)	1 (1.9%)
	*PARP1* (9.6%)	0 (0%)	4 (7.7%)	0 (0%)	1 (1.9%)
	*CHD1* (15.4%)	0 (0%)	3 (5.8%)	2 (3.8%)	3 (5.8%)
	*CHD4* (0%)	0 (0%)	0 (0%)	0 (0%)	0 (0%)
	*RAD50* (3.8%)	0 (0%)	1 (1.9%)	1 (1.9%)	0 (0%)
	*MRE11* (11.5%)	1 (1.9%)	2 (3.8%)	3 (5.8%)	0 (0%)
	*CUL4A* (1.9%)	0 (0%)	0 (0%)	1 (1.9%)	0 (0%)
	*SMARCA4* (5.8%)	1 (1.9%)	1 (1.9%)	1 (1.9%)	0 (0%)
DNA damage checkpoint	*ATM* (19.2%)	1 (1.9%)	5 (9.6%)	3 (5.8%)	1 (1.9%)
	*ATR* (17.3%)	2 (3.8%)	3 (5.8%)	4 (7.7%)	0 (0%)
	*RAD21* (1.9%)	0 (0%)	1 (1.9%)	0 (0%)	0 (0%)
DNA repair	*NBN* (1.9%)	0 (0%)	1 (1.9%)	0 (0%)	0 (0%)
	*PRKDC* (3.8%)	1 (1.9%)	1 (1.9%)	0 (0%)	0 (0%)
	*PIK3CA* (3.8%)	2 (3.8%)	0 (0%)	0 (0%)	0 (0%)
Homologous recombination	*BRCA1* (5.8%)	1 (1.9%)	0 (0%)	1 (1.9%)	1 (1.9%)
	*BRCA2* (21.2%)	1 (1.9%)	6 (11.5%)	1 (1.9%)	3 (5.8%)
	*RAD51* (5.8%)	0 (0%)	1 (1.9%)	1 (1.9%)	1 (1.9%)
	*PALB2* (7.7%)	2 (3.8%)	1 (1.9%)	1 (1.9%)	0 (0%)
DNA mismatch repair	*MSH6* (15.4%)	0 (0%)	4 (7.7%)	1 (1.9%)	3 (5.8%)
	*PMS2* (7.7%)	1 (1.9%)	0 (0%)	1 (1.9%)	2 (3.8%)
	*MLH1* (0%)	0 (0%)	0 (0%)	0 (0%)	0 (0%)
	*MSH2* (0%)	0 (0%)	0 (0%)	0 (0%)	0 (0%)
Histone methyltransferase	*KMT2D* (30.8%)	2 (3.8%)	7 (13.5%)	4 (7.7%)	3 (5.8%)
	*KMT2C* (28.8%)	0 (0%)	5 (9.6%)	4 (7.7%)	6 (11.5%)
	*KDM6A* (5.8%)	0 (0%)	2 (3.8%)	0 (0%)	1 (1.9%)
WNT pathway	*CTNNB1* (15.4%)	7 (13.5%)	1 (1.9%)	0 (0%)	0 (0%)
	*CSNK2B* (1.9%)	1 (1.9%)	0 (0%)	0 (0%)	0 (0%)
SHH pathway	*PTCH1* (11.5%)	0 (0%)	4 (7.7%)	2 (3.8%)	0 (0%)
	*SUFU* (1.9%)	0 (0%)	1 (1.9%)	0 (0%)	0 (0%)
	*SMO* (1.9%)	0 (0%)	0 (0%)	0 (0%)	1 (1.9%)
	*GLI2* (9.6%)	2 (3.8%)	0 (0%)	1 (1.9%)	2 (3.8%)
Other	*TP53* (1.9%)	0 (0%)	1 (1.9%)	0 (0%)	0 (0%)
	*APC* (3.8%)	1 (1.9%)	0 (0%)	0 (0%)	1 (1.9%)
	*DDX3X* (7.7%)	3 (5.8%)	1 (1.9%)	0 (0%)	0 (0%)
	*PTEN* (3.8%)	0 (0%)	2 (3.8%)	0 (0%)	0 (0%)
	*NOTCH1* (17.3%)	3 (5.8%)	2 (3.8%)	2 (3.8%)	2 (3.8%)
	*NOTCH2* (19.2%)	0 (0%)	3 (5.8%)	4 (7.7%)	3 (5.8%)
	*TERT* (3.8%)	0 (0%)	0 (0%)	1 (1.9%)	1 (1.9%)
	*SDHA* (26.9%)	4 (7.7%)	4 (7.7%)	2 (3.8%)	4 (7.7%)
	*NF1* (7.7%)	0 (0%)	2 (3.8%)	1 (1.9%)	1 (1.9%)
	*NF2* (3.8%)	1 (1.9%)	1 (1.9%)	0 (0%)	0 (0%)
	*RB1* (1.9%)	0 (0%)	0 (0%)	0 (0%)	1 (1.9%)
	*RUNX1* (3.8%)	1 (1.9%)	1 (1.9%)	0 (0%)	0 (0%)
	*KRAS* (0%)	0 (0%)	0 (0%)	0 (0%)	0 (0%)
	*RET* (5.8%)	1 (1.9%)	1 (1.9%)	0 (0%)	1 (1.9%)
	*VHL* (1.9%)	1 (1.9%)	0 (0%)	0 (0%)	0 (0%)
	*ALK* (9.6%)	0 (0%)	3 (5.8%)	1 (1.9%)	1 (1.9%)
	*CDH1* (13.5%)	1 (1.9%)	1 (1.9%)	2 (3.8%)	3 (5.8%)
	*NRAS* (1.9%)	0 (0%)	0 (0%)	0 (0%)	1 (1.9%)
	*SDHB* (0%)	0 (0%)	0 (0%)	0 (0%)	0 (0%)

**Table 4 cancers-12-00653-t004:** Clinical parameters, relevant clinical findings, driver mutation, associated somatic mutations, potential genetic predisposition, and mutation frequency in six of ten SHH MBs with WES studies among seventeen SHH MBs in a cohort series of 52 childhood MBs in Taiwan.

Case No.	Age at Dx/Sex	Molecular Subtypes	Histological Phenotype	Metastatic Status at Dx	Relevant Clinical Findings	Driver Mutation	Associated Somatic Mutations	Potential Genetic Predisposition	Mutation Frequency (/Mb) †	Post-Resection Treatment	Outcome/Age at FU
1	8.5 y/F	SHH α	LCA	M3	Early recurrence	*TP53*	*BRCA2*, *SDHA*, *RUNX1*, *ALK*	*TP53* germline mutation (Li-Fraumeni syndrome) §	7.64	RT (CSI + PFI) + CMT	Died at 9.6 y/o from recurrence
2	6.3 y/F	SHH α	DNMB	M0-1	Anaplastic astrocytoma at 8.1 y/o, T-lymphoblastic lymphoma at 11 y/o	*MSH6*	*PRPA1*, *SMARCA4*, *ATM*, *ATR*, *RAD21*, *KMT2D*, *NOTCH1*	Biallelic *MSH6* germline mutation (CMMR-D)	9.23	RT (CSI + PFI) + CMT	Died at 11.6 y/o from progressing anaplastic astrocytoma
3	3.6 y/F	SHH β ‡	Classic	M0-1	Multiple adenomatous hyperplasia and nodular goiter at 3.8 y/o	*PTEN*	*ATM*, *KMT2C*, *SHHA*, *NF2*	*PTEN* germline mutation (PHTS) §	5.62	RT (CSI + PFI)	Died at 11.4 y/o from recurrence
4	5.8 y/F	SHH γ	Classic	M0-1	Parotid rhabdomyosarcoma at 4.3 y/o	*PTCH1*	*CHD1*, *MRE11*, *BRCA2*, *KMT2D*	*PTCH1* somatic mutation (Gorlin syndrome)	5.65	CMT	Alive at 14.3 y/o
5	14.5 y/M	SHH β	Classic	M0-1	Buccal myxofibrosarcoma at 17.2 y/o	*TERT* *	*PARP1*, *KMT2*, *KMT2D*, *NOTCH2*	*TERT* promotor mutation *	7.95	RT (CSI + PFI) + CMT	Alive at 26 y/o
6	2.2 y/M	SHH γ	DNMB	M0-1	No	*SUFU*	*PALB2*, *PTEN*, *SDHA*	*SUFU* and *PALB2* germline mutation (Gorlin syndrome)	15.98	RT (PFI) + CMT	Alive at 12.7 y/o

CMMR-D: Constitutional Mismatch Repair Deficiency, CMT: Chemotherapy, CSI: Craniospinal irradiation, Dx: Diagnosis, FU: Follow-up, LCA: Large-cell/anaplastic, PFI: Posterior fossa irradiation, PHTS: PTEN hamartoma tumor syndrome, RT: Radiotherapy. ^†^ Identified from RNA-Seq. § The relevant clinical findings did not fulfil the diagnostic criteria. ‡ Classification of SHH β MB from recurrent tumor at the age of 19.9 years. * To be confirmed.

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
