# Peer review of "Molecular-Clinical Correlation in Pediatric Medulloblastoma: A Cohort Series Study of 52 Cases in Taiwan"

_cancers, 2020, doi:10.3390/cancers12030653_

Round 1
Reviewer 1 Report
In this manuscript, Wu et al. conducted an in-depth analysis of 52 MB pediatric patients. The authors performed RNA-seq and DNA methylation (Met) analysis of these patients and stratified the patient based the molecular signatures into four previously identified classification groups- WNT, SHH, Group 3, Group 4. Furthermore, using RNA-seq stratification, the authors selected 51 genes and investigated their mutation landscape across the different subgroups. Overall their findings are informative and in-line with previous reports, though they provide lesser of novel findings which is somewhat surprising considering the type of analysis that was performed and the size of the current cohort. Nonetheless, the studies are in principle well-conceived and appropriate controls are included. The analysis and interpretation of the presented data are acceptable; however, I suggest some minor changes to the manuscript to target a broader readership as highlighted below:
1. The manuscript and data are overwhelming, thus I suggest placing less-critical figures to supplemental data, such as Figures 2 F-O, or Figure 3B. Alternatively, the rationale for conducting this type of analysis and insights gained from the data should be better emphasized in the text.
2. It should be better explained why the list of 51-selected genes were selected for mutations calling analysis.
3. The introduction needs to be smoothed out and the rationale of the current study needs to be better explained. In its current format, it reads like a somewhat-confusing mini-review without placing the relevant literature in context of the current work. Furthermore, the four molecular subgroups (WNT, SHH, Group 3, Group 4) should be better explained.
4. It it surprising the fact that some well-known mutations, such as in P53 and NF1, are very low in the current cohort compared with some less-prevalent mutations. Can the authors discuss this?
5. Some parts of the discussion should be integrated into the result section to allow a better understanding of the findings, the rationale of the work, and to keep current findings in the context of work in the field. This should be done for sections 3.1-3.3
Author Response
Response to Reviewer 1 Comments
Comments from Reviewer 1:
In this manuscript, Wu et al. conducted an in-depth analysis of 52 MB pediatric patients. The authors performed RNA-seq and DNA methylation (Met) analysis of these patients and stratified the patient based the molecular signatures into four previously identified classification groups- WNT, SHH, Group 3, Group 4. Furthermore, using RNA-seq stratification, the authors selected 51 genes and investigated their mutation landscape across the different subgroups. Overall their findings are informative and in-line with previous reports, though they provide lesser of novel findings which is somewhat surprising considering the type of analysis that was performed and the size of the current cohort. Nonetheless, the studies are in principle well-conceived and appropriate controls are included. The analysis and interpretation of the presented data are acceptable; however, I suggest some minor changes to the manuscript to target a broader readership as highlighted below:
Point 1: The manuscript and data are overwhelming, thus I suggest placing less-critical figures to supplemental data, such as Figures 2 F-O, or Figure 3B. Alternatively, the rationale for conducting this type of analysis and insights gained from the data should be better emphasized in the text.
Response to Point 1
Thank you for the suggestion:
- Figure 2 F-O and Figure 3B are moved to Supplementary as Figure S2a-i and Figure S6, respectively.
- The rationale for conducting this type of analysis and our purposes were further addressed in the revised Introduction section as suggested.
- The insights gained from the data will be as follows:
- To confirm the prognostic value of extent of resection (subtotal resection) and metastasis at diagnosis.
- To demonstrate the genetic background of extent of resection (subtotal resection) and metastasis at diagnosis across different molecular subgroups.
- To confirm the clinical applicability of the adjusted Heidelberg risk stratification schedule in our cohort series and furthermore for national wide use.
- To identify genes and pathways associated with metastasis or recurrence across different subgroups for risk stratification and therapeutic targeting.
- To identify somatic driver gene mutations and linkage with germline mutation, genetic predisposition, genetic counselling, and personalized care.
- To identify somatic mutations in pathways of DNA damage response (DDR) and cell cycle checkpoint. The results will be the basis of further studies to develop DDR based therapy for synthetic lethality in high risk MB with resistance or metastasis.
These insights gained from the data are addressed and distributed in the sections of Results.
Point 2: It should be better explained why the list of 51-selected genes were selected for mutations calling analysis.
Response to point 2
Thank you for reviewer’s concern and recommendation.
The reasons why the list of 51-selected genes was selected is explained as follows:
- The selection of these genes began from the observation of five patients in this cohort series presenting clinical findings of potential genetic predisposition. The potential relevant driver mutations were TP53, MSH6, PTEN, PTCH, and TERT promoter. Zhang et al. [1] in 2105 reported 21 genes associated with autosomal dominant cancer-predisposition syndrome in pediatric cancer. Waszak et al. [2]in 2018 identified six genes as MB predisposition genes. Merging these three observations, 26 genes with relevance to genetic predisposition in MBs and pediatric cancers were selected as part of the 51-selected genes.
- Within these 26-selected genes, 2 genes (BRCA1, BRCA2) and 4 genes (MSH6, PMS2, MLH1, MSH2) were genes in homologous recombination and DNA mismatch repair, respectively. Pugh et al. in 2012 [3] uncovered subgroup specific somatic mutations and affected pathways of MB that included histone methyl-transferases, chromatin remodeling, WNT pathway, and SHH pathway. The remaining of the 51-selected genes were chosen from pathways of DDR, cell cycle checkpoint, and subgroup specific pathways of MB genesis such as WNT and SHH.
- The explanation of choosing the 51-selected genes for somatic mutation detection was descripted in the Introduction section (Line 142-157, page 3-4).
Point 3: The introduction needs to be smoothed out and the rationale of the current study needs to be better explained. In its current format, it reads like a somewhat-confusing mini-review without placing the relevant literature in context of the current work. Furthermore, the four molecular subgroups (WNT, SHH, Group 3, Group 4) should be better explained.
Response to point 3.
Thank you very much for the advices.
The response to Point 3 is as follows:
- We reorganize the section of Introduction and the rationale of the current study was further addressed.
- In the revision of the Introduction, the relevant literatures are placed in context of the current work in the Introduction section.
- The four molecular subgroups are further descripted in Introduction section (Line 80-93, page 2).
Point 4: It is surprising the fact that some well-known mutations, such as in P53 and NF1, are very low in the current cohort compared with some less-prevalent mutations. Can the authors discuss this?
Response to point 4.
Thank you for highlighted low TP53 and NF1 mutation in our cohort series.
Our response to point 4 is as follows:
- TP53 mutation
In our cohort series of 52 childhood MBs, the prevalence of TP53 mutation was 1.9% (1/52) that distributed in SHH subgroup with mutation rate of 5.9% (1/17). According to the report of Zhukova et al. in 2013, the mutation rates of TP53 were high in WNT (16%) and SHH (21%), but absent in group 3 (0%) and rare in group 4 MB (0.8%) [4]. In this study, the TP53 sequencing was performed on the entire coding sequence (exons 2 through 11) comparing to the identification by RNA-Seq in our cohort series. In the report of Northcott et al. in 2017, similar results were found with TP53 mutation rate of 16% in WNT and 13% in SHH MBs, respectively [5]. Waszak et al. in 2018 [2] identified six genes that included TP53 as MB predisposing genes. This study found TP53 germline mutations in 14(1%) of all 1022 patients with MB and were only identified in SHH MBs. In our cohort study, the only one TP53 mutation distributed in SHH subgroup and presented as somatic and germline mutations. The prevalence of germline mutation of TP53 in our series was 1 (1.9%) of 52 patients.
- TP53 mutations was the most significant risk factor in SHH MBs [4]. Poor survival of non-subgrouped childhood MB with somatic mutations of TP53 was reported [6]. We described our low TP53 mutation in Result section 2.4 (line 283-294, page 9).
- NF1 mutation
The prevalence of NF1 in our series was 4 (7.7%) in 52 patients and the distributions in molecular subgroups were 2 (11.8%) in SHH, 1 (6.7%) in Group 3, and 1 (7.7%) in Group 4 tumors. There were a few documented cases of MBs in patients of neurofibromatosis [7]. Wolfram G et al. in 1995 reported the study of 51 childhood brain tumors for mutations of the FLR-axon within the GAP-related domain of the NF1 gene (NF1-GRD). MBs were part of the 51 tumors. No mutations were found [8].
Point 5: Some parts of the discussion should be integrated into the result section to allow a better understanding of the findings, the rationale of the work, and to keep current findings in the context of work in the field. This should be done for sections 3.1-3.3
Response Point 5.
Thank you for the recommendation.
According to the suggestions of reviewer, the sections 3.1-3.3 are integrated into the Result section as follows:
Section 3.1 to section 2.2 (Line 195-257, page 6-8), Section 3.2 to section 2.5 (Line 298-320, page 9-10), and Section 3.3 to section 2.6 (Line 321-356, page 10-11).
REFERENCES
- Zhang, J.; Walsh, M.F.; Wu, G.; Edmonson, M.N.; Gruber, T.A.; Easton, J.; Hedges, D.; Ma, X.; Zhou, X.; Yergeau, D.A., et al. Germline Mutations in Predisposition Genes in Pediatric Cancer. The New England journal of medicine 2015, 373, 2336-2346, doi:10.1056/NEJMoa1508054.
- Waszak, S.M.; Northcott, P.A.; Buchhalter, I.; Robinson, G.W.; Sutter, C.; Groebner, S.; Grund, K.B.; Brugieres, L.; Jones, D.T.W.; Pajtler, K.W., et al. Spectrum and prevalence of genetic predisposition in medulloblastoma: a retrospective genetic study and prospective validation in a clinical trial cohort. The Lancet. Oncology 2018, 19, 785-798, doi:10.1016/s1470-2045(18)30242-0.
- Pugh, T.J.; Weeraratne, S.D.; Archer, T.C.; Pomeranz Krummel, D.A.; Auclair, D.; Bochicchio, J.; Carneiro, M.O.; Carter, S.L.; Cibulskis, K.; Erlich, R.L., et al. Medulloblastoma exome sequencing uncovers subtype-specific somatic mutations. Nature 2012, 488, 106-110, doi:10.1038/nature11329.
- Zhukova, N.; Ramaswamy, V.; Remke, M.; Pfaff, E.; Shih, D.J.; Martin, D.C.; Castelo-Branco, P.; Baskin, B.; Ray, P.N.; Bouffet, E., et al. Subgroup-specific prognostic implications of TP53 mutation in medulloblastoma. Journal of clinical oncology : official journal of the American Society of Clinical Oncology 2013, 31, 2927-2935, doi:10.1200/jco.2012.48.5052.
- Northcott, P.A.; Buchhalter, I.; Morrissy, A.S.; Hovestadt, V.; Weischenfeldt, J.; Ehrenberger, T.; Grobner, S.; Segura-Wang, M.; Zichner, T.; Rudneva, V.A., et al. The whole-genome landscape of medulloblastoma subtypes. Nature 2017, 547, 311-317, doi:10.1038/nature22973.
- Tabori, U.; Baskin, B.; Shago, M.; Alon, N.; Taylor, M.D.; Ray, P.N.; Bouffet, E.; Malkin, D.; Hawkins, C. Universal poor survival in children with medulloblastoma harboring somatic TP53 mutations. Journal of clinical oncology : official journal of the American Society of Clinical Oncology 2010, 28, 1345-1350, doi:10.1200/jco.2009.23.5952.
- Rosenfeld, A.; Listernick, R.; Charrow, J.; Goldman, S. Neurofibromatosis type 1 and high-grade tumors of the central nervous system. Child's Nervous System 2010, 26, 663-667, doi:10.1007/s00381-009-1024-2.
- Scheurlen, W.G.; Senf, L. Analysis of the GAP-related domain of the neurofibromatosis type 1 (NF1) gene in childhood brain tumors. Int J Cancer 1995, 64, 234-238, doi:10.1002/ijc.2910640404.
Reviewer 2 Report
In this manuscript the authors utilized 52 frozen tumor tissues of childhood Medulloblastoma (MB) and aimed to identify a molecular-clinical correlation and somatic mutation for exploring risk-adapted treatment, drug targets, and potential genetic predisposition. The authors did a thorough study, generating data from RNA seq and DNA methylation array and performed molecular subgrouping and clinical correlation analysis. The results presented in the paper provide valuable information for updated risk adapted treatment and are potentially of interest to the readership of Cancers Journal. Minor comments: figure 1 and 2 in the result section (2.2 and 2.3) are not arranged in order. It will be helpful for the readers if they are written in the same order as they presented the figures in the article.
Specific comments:
- Figure 4a, detection of somatic mutations in 51 selected genes and affected pathways from RNA-seq data could be very useful for assessing targeted therapy. However, it is important to further analyze whether expression of top 20 genes that are up or downregulated (RNA seq analysis in Fig. 2a) in each molecular subtype (WNT, SHH, group 3 and group 4) are positively or negatively correlated with mutation accumulation in the tumors.
- Identifications of somatic mutations in DNA damage repair genes involved in single-strand breaks repair pathway (PARP1) or double strand breaks repair pathways (BRAC1/2) and mismatch repair (MMR) (MSH6/PMS2) are significant observations. This may raise the possibility that either targeting one pathway with inhibitors can induce synthetic lethality in some set of tumors. Synthetic lethality approach through PARP1- inhibitor, olaparib, has been successful to some extent in breast and ovarian cancer with BRAC1/2 mutations, however, this approach is unexplored and still need to be validated in MB. Therefore, this study can open up a new opportunity for testing potential therapeutic benefits of using DNA repair pathways inhibitors.
- Figure 2e, The SHH subgroup could be further subtyped as SHH a, SHH b and SHH g. Clustering of the gene expression from RNA seq data by T-distributed Stochastic Neighbor Embedding (t-SNE) analysis revealed they have three distinct gene expression profile (Fig. 2e). However, it is not clear how many common genes were identified in SHH group during molecular subtyping as shown in Figure 2a.
- Even though the patients’ number is low for each SHH subgroups, are their expression profiles correlated with disease recurrence or can be used as for therapy response or clinical risk stratifications?
Author Response
Response to Reviewer 2 Comments
Comments from Reviewer 2:
In this manuscript the authors utilized 52 frozen tumor tissues of childhood Medulloblastoma (MB) and aimed to identify a molecular-clinical correlation and somatic mutation for exploring risk-adapted treatment, drug targets, and potential genetic predisposition. The authors did a thorough study, generating data from RNA seq and DNA methylation array and performed molecular subgrouping and clinical correlation analysis. The results presented in the paper provide valuable information for updated risk adapted treatment and are potentially of interest to the readership of Cancers Journal. Minor comments: figure 1 and 2 in the result section (2.2 and 2.3) are not arranged in order. It will be helpful for the readers if they are written in the same order as they presented the figures in the article.
Response to comments of Reviewer
Thank you for the comment.
To follow the recommendation of Point 1 from reviewer 1, we move Figure 2 F-O to Figure S2a-i. After that, section (2.2 and 2.3) are also revised to keep writing in the same order as they presented in figures in the article as recommended.
Point 1: Figure 4a, detection of somatic mutations in 51 selected genes and affected pathways from RNA-seq data could be very useful for assessing targeted therapy. However, it is important to further analyze whether expression of top 20 genes that are up or downregulated (RNA seq analysis in Fig. 2a) in each molecular subtype (WNT, SHH, group 3 and group 4) are positively or negatively correlated with mutation accumulation in the tumors.
Response to Point 1:
Thank you for the recommendation.
The response to Point 1 is as follows:
- According to the recommendation, we further analyzed the top 20 genes that are up or downregulated (RNA seq analysis in Fig. 2a) across the four molecular subgroups (WNT, SHH, group 3 and group 4) for positive or negative correlation with mutation accumulation in the tumors (Figure S7). The correlation coefficient and p-value were summarized in Table S1. Interestingly, we found that in the SHH group, most of the top 20 upregulated genes have positive associations with the mutation numbers, while all the top 20 downregulated genes have negative associations. On the contrary, in Group 4, all the top 20 upregulated genes have negative associations with the mutation numbers, while all the top 20 downregulated genes have positive associations. This reflects the distinct molecular mechanism of the two subtypes, offering novel angles of future investigations. To our knowledge, this phenomenon has not been reported before.
Figure S7. The correlation coefficient of top 20 up- and down-regulated genes in WNT (a, b), SHH (c, d), Group 3 (e, f), and 4 (g, h) and tumor mutation burden.
Table S1. The correlation coefficient of the top 20 upregulated and downregulated genes in the four core subgroups of MB and tumor mutation burden
|
Subgroup |
Expression |
correlation (range) |
p-value (range) |
|
WNT |
up |
-0.119~0.219 |
0.119~0.980 |
|
WNT |
down |
-0.310~0.220 |
0.025~0.999 |
|
SHH |
up |
-0.069~0.376 |
0.006~0.944 |
|
SHH |
down |
-0.405~-0.022 |
0.003~0.877 |
|
G3 |
up |
-0.322~0.247 |
0.020~0.982 |
|
G3 |
down |
-0.202~0.243 |
0.083~0.987 |
|
G4 |
up |
-0.444~-0.035 |
0.001~0.806 |
|
G4 |
down |
0.051~0.437 |
0.001~0.721 |
- The result is described in Result section 2.6 (line 348-356, page 10-11).
Point 2: Identifications of somatic mutations in DNA damage repair genes involved in single-strand breaks repair pathway (PARP1) or double strand breaks repair pathways (BRAC1/2) and mismatch repair (MMR) (MSH6/PMS2) are significant observations. This may raise the possibility that either targeting one pathway with inhibitors can induce synthetic lethality in some set of tumors. Synthetic lethality approach through PARP1- inhibitor, olaparib, has been successful to some extent in breast and ovarian cancer with BRAC1/2 mutations, however, this approach is unexplored and still need to be validated in MB. Therefore, this study can open up a new opportunity for testing potential therapeutic benefits of using DNA repair pathways inhibitors.
Response to Point 2:
Thank you for the consideration and initiation.
The response to Point 2 is as follows:
- Synthetic lethality is potentially exploitable by using DDR inhibitor in childhood MB presenting DDR dysregulation. Upregulated DDR pathways confer to resistance of DNA damage therapy and DDR inhibitor may overcome resistance [1]. DNA repair/cell cycle checkpoint genes including PARP1, BRCA1, ATM and TP53 may also be associated with tumor metastasis [2].
- TP53 was the most significant risk factor in SHH MBs. On the other hand, incidence of metastasis of MB by subgroup varies from 17.9% in WNT, 19.1% in SHH, 46.5% in group3, and 29.7% in group 4 tumors in the report of Northcott et al. in 2011 [3]. Metastasis is the most significant prognostic factor of non-WNT childhood MBs. PARP1 inhibitor may be a promising treatment for some MB with DDR dysregulation or for metastatic or resistant MBs. Further preclinical study in metastatic MB cell lines and mice model is required.
- We addressed Point 2 in Discussion section 3.1 (line 419-428, page 15-16).
Point 3: Figure 2e, The SHH subgroup could be further subtyped as SHH a, SHH b and SHH g. Clustering of the gene expression from RNA seq data by T-distributed Stochastic Neighbor Embedding (t-SNE) analysis revealed they have three distinct gene expression profile (Fig. 2e). However, it is not clear how many common genes were identified in SHH group during molecular subtyping as shown in Figure 2a.
Response to Point 3:
Thanks for the question. The response to point 3 is as follows:
- In this cohort series, subtype clustering of SHH α, β, and γ was performed based on the top 1% of the most differentially expressed genes (n=201) and probes list from a study [4]. These 201 genes were adopted from the report of Cavalli et al. in 2017 [4] for SHH subtyping.
- To answer the question of Point 3, we analyze the expression profiles of the top and bottom 20 genes in the three SHH subtypes. The expression profiles of these genes are shown in Figure S1a, b. In heatmap of the bottom 20 genes, common genes were found across SHH subtypes.
- The result of Point 3 was described in Result section 2.2 (Line 205-206, page 7).
Figure S1. Heatmap of top (a) and bottom (b) 20 most differentially expressed genes in the three SHH subtype of the cohort series.
Point 4: Even though the patients’ number is low for each SHH subgroups, are their expression profiles correlated with disease recurrence or can be used as for therapy response or clinical risk stratifications?
Response to Point 4.
Thank for this valuable question.
The response to Point 4 is as follows:
- We further perform comparative top 20 metastasis-associated gene expression profiling in tumors with or without recurrence across molecular subgroups (Figure S4a-g). The expression pattern of metastasis-related genes is upregulated in tumors with recurrence in the three SHH subtypes and other molecular subgroups.
- Selected genes of the top 20 upregulated recurrence-associated genes and pathways may work as significant prognosticating factors in each SHH subtypes or other molecular subgroups. Some of these upregulated genes or pathways may have potential of developing targeted therapy. However, further translational research and preclinical testing for highly selected genes/pathways are required.
- The result of Point 4 was descripted in Result section 2.2 (Line 225-227, page 7).
Figure S4. Heatmap of top 20 highly expressed metastasis-associated genes in tumors with recurrence in WNT (a), SHH α (b), SHH β (c), SHH γ (d), Group 3 (e), and Group 4 (f). (g) Gene set enrichment analysis of pathways by top 20 metastasis-associated genes in the three SHH subtypes and other molecular subgroups.
REFERENCES
- Taylor, M.D.; Northcott, P.A.; Korshunov, A.; Remke, M.; Cho, Y.J.; Clifford, S.C.; Eberhart, C.G.; Parsons, D.W.; Rutkowski, S.; Gajjar, A., et al. Molecular subgroups of medulloblastoma: the current consensus. Acta Neuropathol 2012, 123, 465-472, doi:10.1007/s00401-011-0922-z.
- Broustas, C.G.; Lieberman, H.B. DNA damage response genes and the development of cancer metastasis. Radiat Res 2014, 181, 111-130, doi:10.1667/RR13515.1.
- Northcott, P.A.; Korshunov, A.; Witt, H.; Hielscher, T.; Eberhart, C.G.; Mack, S.; Bouffet, E.; Clifford, S.C.; Hawkins, C.E.; French, P., et al. Medulloblastoma comprises four distinct molecular variants. Journal of clinical oncology : official journal of the American Society of Clinical Oncology 2011, 29, 1408-1414, doi:10.1200/JCO.2009.27.4324.
- Cavalli, F.M.G.; Remke, M.; Rampasek, L.; Peacock, J.; Shih, D.J.H.; Luu, B.; Garzia, L.; Torchia, J.; Nor, C.; Morrissy, A.S., et al. Intertumoral Heterogeneity within Medulloblastoma Subgroups. Cancer cell 2017, 31, 737-754.e736, doi:10.1016/j.ccell.2017.05.005.
Reviewer 3 Report
The authors have presented a comprehensive genomic analysis of 52 paediatric medulloblastoma cases in Taiwan. Based on RNAseq and DNA methylation analysis samples were defined into the key 4 MB subgroups. Whole exome sequencing was performed in 10 SHH samples with 5 exhibiting germline aberrations. Overall it is an interesting and well-presented study. Some of the key highlights are attempts to recognize players that could be targeted as well as the presence of germline aberrations in the SHH subgroup.
Specific questions:
It is not clear how the authors chose the panel of 51 clinically relevant genes. Could they elaborate further?
Why did the authors perform WES only in SHH samples and not the rest since Gp3 and Gp4 are considered more heterogeneous?
Again germline information on all samples would have been fantastic to have instead of the focused 10 SHH samples.
Given that some of the patients exhibited recurrence it would have been extremely valuable to understand the mechanisms behind recurrence and resistance. Is there any information in terms of targeting specific genes/pathways from RNAseq data between samples with metastatic potential compared and those with not?
In regards to MB genesis, authors should compare their findings with Hovestadt et al 2019, Nature 572(7767):74-79.
Author Response
Response to Reviewer 3 Comments
Comments from Reviewer 3:
The authors have presented a comprehensive genomic analysis of 52 paediatric medulloblastoma cases in Taiwan. Based on RNAseq and DNA methylation analysis samples were defined into the key 4 MB subgroups. Whole exome sequencing was performed in 10 SHH samples with 5 exhibiting germline aberrations. Overall it is an interesting and well-presented study. Some of the key highlights are attempts to recognize players that could be targeted as well as the presence of germline aberrations in the SHH subgroup.
Specific questions:
Point 1: It is not clear how the authors chose the panel of 51 clinically relevant genes. Could they elaborate further?
Response to Point 1:
Thank you for the positive comments of reviewer and the question in Point 1. This is the same question concerned by reviewer 1 in Point 2.
The reasons why the list of 51-selected genes was selected is explained as follows:
- The selection of these genes began from the observation of five patients in this cohort series presenting clinical findings of potential genetic predisposition. The potential relevant driver mutations were TP53, MSH6, PTEN, PTCH, and TERT promoter. Zhang et al. [1] in 2105 reported 21 genes associated with autosomal dominant cancer-predisposition syndrome in pediatric cancer. Wazak et al. [2] in 2018 identified six genes as MB predisposition genes. Merging these three observations, 26 genes with relevance to genetic predisposition in MBs and pediatric cancers were selected as part of the 51-slected genes.
- Within these 26-selected genes, 2 genes (BRCA1, BRCA2) and 4 genes (MSH6, PMS2, MLH1, MSH2) were genes in homologous recombination and DNA mismatch repair, respectively. Pugh et al. in 2012 [3] uncovered subgroup specific somatic mutations and affected pathways of MB that included histone methyl-transferases, chromatin remodeling, WNT pathway, and SHH pathway. The remaining of the 51-seleced genes were chose from pathways of DDR, cell cycle checkpoint, and subgroup specific pathways of MB genesis such as WNT and SHH.
- The explanation of choosing the 51-selected genes for somatic mutation detection in the was descripted in Introduction Section (Line 142-157, page 3-4).
Point 2: Why did the authors perform WES only in SHH samples and not the rest since Gp3 and Gp4 are considered more heterogeneous?
Point 3: Again germline information on all samples would have been fantastic to have instead of the focused 10 SHH samples.
Response to Point 2 and Point 3.
Thank you for the question and recommendation of Point 2 and 3. The response of Point 2 and 3 is as follows:
- We first generated RNA-Seq and DNA methylation for the purposes of molecular classification, detection of subgroup specific CNVs, and identification of somatic mutations in a panel of 51-selected genes. Subgroup-specific somatic mutations were observed and driver mutations associated with genetic predisposition clustered in SHH MBs. These findings matched with the observation of Waszak et al. in 2018 that the prevalence of genetic predisposition in MB distributed exclusively in SHH subgroups [2].
- Because of limited funding for molecular sequencing in this project, we decided to focus on performing WES through available tumor tissues and blood samples in SHH subgroup for correlating with RNA-Seq detected somatic mutation and the detection of germline mutations in SHH MBs.
- We pointed out this decision in the Introduction section (line 120-122, page 3).
Point 4: Given that some of the patients exhibited recurrence it would have been extremely valuable to understand the mechanisms behind recurrence and resistance. Is there any information in terms of targeting specific genes/pathways from RNA-seq data between samples with metastatic potential compared and those with not?
Response to Point 4:
Thank you for this valuable question. A similar question is asked by reviewer 2 Point 4 in SHH tumors with recurrence.
The response to Point 4 is as follows:
- We further perform comparative top 20 metastasis-associated gene expression profiling in tumors with or without metastasis (Figure S3) and also in tumors with or without recurrence (Figure S4) across molecular subgroups. The expression pattern of metastasis-related genes/pathways is upregulated in tumors with metastasis and in tumors with recurrence in different molecular subgroups. There are overlapping of metastasis-associated genes in each molecular subgroup (Figure S5).
- Selected panel of the top 20 upregulated metastasis-associated genes, recurrence-associated genes and pathways, and also the overlapping metastasis-associated genes/pathways may contribute to therapeutic targets. However, further translational research and preclinical testing for highly selected genes/pathways are required.
- The answer to Point 4 is descripted in Result section 2.2 (Line 225-237, page 7).
Figure S3. Heatmap of top 20 highly expressed metastasis-associated genes in tumors with metastasis in SHH α (a), SHH β (b), Group 3 (c), and Group 4 (d). (e) Gene set enrichment analysis of pathway in metastatic tumors of MB subgroups.
Figure S4. Heatmap of top 20 highly expressed metastasis-associated genes and gene set enrichment analysis in tumors with recurrence in WNT (a), SHH α (b), SHH β (c), SHH γ (d), Group 3 (e), and Group 4 (f). (g) Gene set enrichment analysis of signature pathway in tumors with recurrence in MB subgroups.
Figure S5. The overlapping genes in tumors with metastatic and tumors with recurrent tumor in SHH α (a), Group 3 (b), and Group 4 (c).
Point 5: In regards to MB genesis, authors should compare their findings with Hovestadt et al 2019, Nature 572(7767):74-79.
Response to Point 5:
Thank you very much for the very important suggestion for comparing our findings with the report of Hovestadt et al 2019 regarding MB genesis.
To response to Point 5,
- Hovestadt et al. used single-cell transcriptomics to investigate intra- and inter-tumoral heterogeneity MBs across the four distinct molecular subgroups [4]. The authors found that WNT, SHH, and Group 3 tumors comprised subgroup-specific undifferentiated and differentiated neuronal-like malignant populations. Comparing with Group3, Group 4 tumors were exclusive comprised of differentiated neuronal-like neoplastic cells. In cross-species transcriptional analysis, significant correlations between SHH MB and GNPCs. UBC and GluCN were identified as cellular correlation of Group 4 MBs. Both WNT and Groups 3 may have an extracerebellar origin.
- Molecular classification and related molecular analysis of childhood MB is our very first experience. In this paper, we try to correlate their findings and our clinical observations. Their findings match with our observations as follows:
i). The heterogeneity of gene expression among molecular subgroups and the subgroups-specific expression profile of metastatic-associated genes/pathways in tumor metastasis or tumor recurrence may imply difference of cell of origin in MB genesis and the proportion of undifferentiated and differentiated neuronal-like populations in different molecular subgroups.
- ii) The survival of Group 4 is better than Group 3 MBs observed in the study of Shih et al. [5] and our cohort series. This finding may correlate with the high proportion of differentiated neuronal-like neoplastic cells in Group 4 MB.
iii). We observed tumor metastasis at third ventricular infundibulum recess that occurred in both SHH and Group 4 MB and was not observed in WNT and Group 3 tumors. This finding may due to similar cell-of-origin of SHH and Group (the GNPCs in SHH and the UBC and GluCN cells in Group 4).
- We add description for clinical correlation with the findings of Hovestadt et al in Result section 2.2 (line 238-239, page 7).
REFERENCES
- Zhang, J.; Walsh, M.F.; Wu, G.; Edmonson, M.N.; Gruber, T.A.; Easton, J.; Hedges, D.; Ma, X.; Zhou, X.; Yergeau, D.A., et al. Germline Mutations in Predisposition Genes in Pediatric Cancer. The New England journal of medicine 2015, 373, 2336-2346, doi:10.1056/NEJMoa1508054.
- Waszak, S.M.; Northcott, P.A.; Buchhalter, I.; Robinson, G.W.; Sutter, C.; Groebner, S.; Grund, K.B.; Brugieres, L.; Jones, D.T.W.; Pajtler, K.W., et al. Spectrum and prevalence of genetic predisposition in medulloblastoma: a retrospective genetic study and prospective validation in a clinical trial cohort. The Lancet. Oncology 2018, 19, 785-798, doi:10.1016/s1470-2045(18)30242-0.
- Pugh, T.J.; Weeraratne, S.D.; Archer, T.C.; Pomeranz Krummel, D.A.; Auclair, D.; Bochicchio, J.; Carneiro, M.O.; Carter, S.L.; Cibulskis, K.; Erlich, R.L., et al. Medulloblastoma exome sequencing uncovers subtype-specific somatic mutations. Nature 2012, 488, 106-110, doi:10.1038/nature11329.
- Hovestadt, V.; Smith, K.S.; Bihannic, L.; Filbin, M.G.; Shaw, M.L.; Baumgartner, A.; DeWitt, J.C.; Groves, A.; Mayr, L.; Weisman, H.R., et al. Resolving medulloblastoma cellular architecture by single-cell genomics. Nature 2019, 572, 74-79, doi:10.1038/s41586-019-1434-6.
- Shih, D.J.; Northcott, P.A.; Remke, M.; Korshunov, A.; Ramaswamy, V.; Kool, M.; Luu, B.; Yao, Y.; Wang, X.; Dubuc, A.M., et al. Cytogenetic prognostication within medulloblastoma subgroups. Journal of clinical oncology : official journal of the American Society of Clinical Oncology 2014, 32, 886-896, doi:10.1200/jco.2013.50.9539.